# Insulin Elevates ID2 Expression in Trophoblasts and Aggravates Preeclampsia in Obese ASB4-Null Mice

**DOI:** 10.3390/ijms24032149

**Published:** 2023-01-21

**Authors:** Yukako Kayashima, W. H. Davin Townley-Tilson, Neeta L. Vora, Kim Boggess, Jonathon W. Homeister, Nobuyo Maeda-Smithies, Feng Li

**Affiliations:** 1Department of Pathology and Laboratory Medicine, The University of North Carolina, Chapel Hill, NC 27599, USA; 2Department of Obstetrics and Gynecology, Division of Maternal Fetal Medicine, The University of North Carolina, Chapel Hill, NC 27599, USA

**Keywords:** preeclampsia, maternal obesity, insulin, ID2, trophoblast differentiation

## Abstract

Obesity is a risk factor for preeclampsia. We investigated how obesity influences preeclampsia in mice lacking ankyrin-repeat-and-SOCS-box-containing-protein 4 (ASB4), which promotes trophoblast differentiation via degrading the inhibitor of DNA-binding protein 2 (ID2). *Asb4^−/−^* mice on normal chow (NC) develop mild preeclampsia-like phenotypes during pregnancy, including hypertension, proteinuria, and reduced litter size. Wild-type (WT) and *Asb4^−/−^* females were placed on a high-fat diet (HFD) starting at weaning. At the age of 8–9 weeks, they were mated with WT or *Asb4^−/−^* males, and preeclamptic phenotypes were assessed. HFD-WT dams had no obvious adverse outcomes of pregnancy. In contrast, HFD-*Asb4*^−/−^ dams had significantly more severe preeclampsia-like phenotypes compared to NC-*Asb4^−/−^* dams. The HFD increased white fat weights and plasma leptin and insulin levels in *Asb4^−/−^* females. In the HFD-*Asb4^−/−^* placenta, ID2 amounts doubled without changing the transcript levels, indicating that insulin likely increases ID2 at a level of post-transcription. In human first-trimester trophoblast HTR8/SVneo cells, exposure to insulin, but not to leptin, led to a significant increase in ID2. HFD-induced obesity markedly worsens the preeclampsia-like phenotypes in the absence of ASB4. Our data indicate that hyperinsulinemia perturbs the timely removal of ID2 and interferes with proper trophoblast differentiation, contributing to enhanced preeclampsia.

## 1. Introduction

Preeclampsia is a pregnancy-related disorder, characterized as hypertension, proteinuria, and end-organ damage. It affects 5–8% of all pregnancies and is a leading cause of maternal and neonatal mortality and morbidity [1]. The pathogenesis of preeclampsia is believed to originate in the placenta, with alterations in trophoblast function, resulting in insufficient placental perfusion and release of pro-inflammatory and anti-angiogenic factors into the maternal circulation [2]. 

Maternal obesity is a major risk factor for preeclampsia, and the increase in the prevalence of preeclampsia mirrors an increase in obesity in the U.S. [3,4] and in other countries [5,6]. The mechanism whereby maternal obesity influences preeclampsia is unknown, although obesity-related inflammation, oxidative stress, insulin resistance, and underlying hypertension have been suggested [7]. However, at what stage(s) and how these adverse effects of maternal obesity influence the development of preeclampsia are incompletely understood. In this study, we test our hypothesis that the adverse effects of maternal obesity on preeclampsia start early at the trophoblast differentiation stage, using a mouse model of preeclampsia lacking ankiryn-repeat-and-suppressor of cytokine signaling (SOCS)-box-containing-protein 4 (ASB4). 

ASB4 is a component of protein ubiquitin ligase complexes and provides target substrate recognition for ubiquitination and subsequent proteasomal degradation [8]. It is known to contribute to the differentiation of trophoblast stem cells into the giant trophoblasts necessary for embryo implantation. It also promotes the differentiation of stem cells into vascular cells during placental development and embryogenesis in mice [8,9]. Its spatial-temporal expression is tightly regulated during embryo development. Its expression is low in E7.5 embryos but is quickly induced in E8.5, peaking at E9.5 and then diminishing by E10.5, and is limited to a small subset of cells in adults [9]. Similarly, its expression is turned on 8.5 dpc and peaks in 9.5 dpc in the placenta but is rapidly diminished by 13.5 dpc [9]. Placentas that lack ASB4 display immature vascular patterning and retain the expression of placental progenitor markers, including the inhibitor of DNA-binding protein 2 (ID2) [8]. ID2 is a member of the anti-differentiation ID protein family, which shares significant structural similarity to the basic helix-loop-helix (bHLH) family of transcription factors but lacks the basic domain [10]. Through heterodimerization with functional factors, ID2 blocks the transcription of pro-differentiation elements by preventing bHLH dimerization and subsequent translocation into the nucleus [11]. ASB4 mediates the ubiquitination and proteasomal degradation of ID2, and these processes are essential for the differentiation of cells [8]. ASB4 is thus a key element for trophoblast differentiation into vascularization. However, it is not the sole mechanism for the removal of ID2 and cell differentiation, since placentation, although perturbed, continues in its absence [8]. *Asb4^−/−^* female mice develop mild PE-like phenotypes during pregnancy, including increased systolic blood pressure (SBP) and urinary albumin excretion and decreased litter size [8,12]. Here, by feeding an HFD or regular normal chow (NC) to *Asb4^−/−^* females before and through the entire pregnancy, we demonstrate that HFD-induced maternal obesity significantly worsens all the preeclampsia-like phenotypes in *Asb4^−/−^* dams. HFD-*Asb4^−/−^* dams had increased plasma insulin and leptin levels, and placentas from these dams had sustained ID2 expression to later stages. In contrast, the same HFD treatment regimen had no obvious adverse effect on WT dams. By exposing human first-trimester trophoblasts to obesity-related plasma factors, we demonstrate that insulin has a direct effect on the regulation of ID2 levels, suggesting that the adverse effects of maternal obesity on preeclampsia begin at an early stage of placental development.

## 2. Results

### 2.1. Asb4^−/−^ Female Mice Have More Fat Than WT Female Mice and a High-Fat Diet (HFD) Exacerbates It 

At the time of weaning (~23 days old)**,** the body weight (BW) of *Asb4^−/−^* female mice was slightly higher than that of WT females (*Asb4^−/−^*: 11.1 ± 0.1 g vs. WT:10.5 ± 0.2 g, *n* = 7–15, *p* = 0.06). Five weeks later, by the time of mating, the BW difference between WT and *Asb4^−/−^* females on NC reached a significant level (Appendix A). HFD feeding further increased their BW in an additive fashion to the age. Among the dams at 18.5 dpc, however, two-way ANOVA showed that lacking ASB4 significantly increases the body weight of dams (Appendix A). 

Both genotype and diet significantly affected the visceral and subcutaneous fat weights of pregnant females, while brown fat was affected by the ASB4 genotype but not by diet (Figure 1A, Appendix A). After approximately 8 weeks on different diets, non-pregnant HFD-*Asb4^−/−^* females acquired significantly more visceral fat than non-pregnant NC-*Asb4^−/−^* dams (Appendix A). Two-way ANOVA showed that HFD is a factor in increasing visceral and subcutaneous fat but not brown fat. Pregnancy significantly affected the body fat distribution in *Asb4^−/−^* dams independently of diet. While it increased subcutaneous fat and brown fat weights, visceral fat weight was significantly less in pregnant females (Appendix A).

### 2.2. A High-Fat Diet (HFD) Increases Circulating Levels of Cholesterol, Insulin, and Leptin in Asb4^−/−^ Dams without Changes in Glucose Levels

HFD-*Asb4^−/−^* dams had ~1.6× higher plasma cholesterol levels than NC-*Asb4^−/−^* dams. (Figure 1B). Two-way ANOVA revealed significant effects of diet but not of the genotype on the plasma cholesterol levels in pregnant mice (Appendix A). The plasma levels of triglyceride and glucose were not different among the four groups of dams (Figure 1C,D). Among *Asb4^−/−^* females, pregnancy significantly reduced plasma cholesterol and glucose levels, while pregnancy had an increasing effect on plasma triglyceride levels (Appendix A). 

We found ~3× higher plasma insulin levels in HFD-*Asb4^−/−^* dams than in NC-*Asb4^−/−^* dams. The HFD did not alter the plasma insulin levels in WT dams (Figure 1E). Two-way ANOVA among pregnant mice showed that both the *Asb4* genotype and HFD significantly increased plasma insulin levels, with a statistically significant interaction between the effects of the genotype and diet, revealing the exaggerated increase in plasma insulin levels uniquely in the HFD-*Asb4^−/−^* dams (Appendix A). Elevated plasma insulin levels but unaltered glucose levels suggest that the HFD-*Asb4^−/−^* dams developed insulin resistance. In contrast, analysis among *Asb4^−/−^* female mice showed that HFD but not pregnancy had an effect on the plasma insulin levels (Appendix A).

Leptin, a metabolic factor that increases in obesity, has been reported to be involved in preeclampsia [13,14]. Plasma leptin levels were higher in HFD-*Asb4^−/−^* females than in NC-*Asb4^−/−^* females at mating and 18.5 dpc, although the difference between the two groups became smaller at the later gestational stage (Appendix A). Because mouse placentas do not synthesize significant amounts of leptin unlike humans or rats [15], we measured the mRNA of *Lpn* (a gene encoding leptin) in gonadal adipose tissue at 18.5 dpc. The mRNA level of *Lpn* was not different between the two groups (Appendix A). The morphology of the gonadal adipose tissue did not show obvious differences between the *Asb4^−/−^* dams fed the different diets (Appendix A).

### 2.3. HFD-Asb4^−/−^ Dams Show Aggravated Preeclampsia-like Phenotypes

At 13.5 dpc, the total number of fetuses (both alive and dead) was not different among the four groups of dams (Appendix A), but HFD-*Asb4^−/−^* dams had more reabsorbed fetuses compared to NC-*Asb4^−/−^* dams. WT dams did not have reabsorbed fetuses regardless of the diet. 

At 18.5 dpc, NC-*Asb4^−/−^* dams had on average 5.0 surviving fetuses, which was less than NC-WT dams (7.0 fetuses), but the difference did not reach a significant level (Figure 2A). In contrast, HFD-*Asb4^−/−^* dams had a markedly smaller number of surviving fetuses (2.6 per pregnancy) on average (Figure 2A). The average weight of the surviving fetuses was not different among the four groups (Figure 2B). However, *Asb4^−/−^* dams had higher placental weights than WT dams regardless of diet (Figure 2C). *Asb4^−/−^* dams had a lower fetal weight/placental weight (F/P) ratio (Figure 2D), which is consistent with previous reports that preeclamptic mothers have smaller F/P ratios [16,17]. Thus HFD-*Asb4^−/−^* dams showed poor pregnancy outcomes.

The HFD treatment regimen we applied in this study did not cause an obvious detrimental effect on maternal proteinuria in WT dams (urinary albumin/creatinine ratio (ACR): 0.04 ± 0.01 mg/mg in WT NC vs. 0.07 ± 0.03 mg/mg in WT HFD, *p* = 0.62). Collectively, the data demonstrated that an HFD does not have adverse effects on WT dams; therefore, we focused on comparing *Asb4^−/−^* females on different diets from now on. Among *Asb4^−/−^* females, 8 weeks of HFD feeding did not alter the systolic blood pressure (SBP) in virgin *Asb4^−/−^* female mice. The SBP of NC-*Asb4^−/−^* dams was 12.5 mmHg higher than their non-pregnant counterparts, while the SBP of HFD-*Asb4^−/−^* dams was 22 mmHg higher than their non-pregnant counterparts (Figure 3A). The ACR was about 1.9× higher in HFD-*Asb4^−/−^* dams than in NC-*Asb4^−/−^* dams (Figure 3B). Glomerular endotheliosis is the hallmark of the kidney lesion of preeclampsia. In NC-*Asb4^−/−^* dams, about 10% of glomeruli were injured, and the injury rate increased to about 48% in HFD-*Asb4^−/−^* dams (Figure 3D). 

Taken together, we concluded that HFD-*Asb4^−/−^* dams had more severer preeclampsia-like symptoms than NC-*Asb4^−/−^* dams. 

### 2.4. HFD-Asb4^−/−^ Dams Have Decreased Vascular Endothelial Growth Factor (VEGF) Levels

Previously, we reported that *Asb4^−/−^* dams had lower circulating VEGF levels than WT dams [12], and we and others proposed that lower VEGF levels contribute to preeclampsia-like symptoms in pregnant mice [18,19]. Altered VEGF levels associated with obesity play a role in the endothelial dysfunction of preeclampsia [7]. Therefore, we measured the plasma VEGF levels in *Asb4^−/−^* dams on different diets. The plasma levels of VEGF were significantly lower in HFD-*Asb4^−/−^* dams than those in NC-*Asb4^−/−^* dams (Figure 4A and Appendix A). The levels of VEGF in adipose tissues and the placenta were also determined because these two tissues are the major source of VEGF in pregnancies with obesity [7]. The VEGF mRNA and protein levels in the placentas from HFD-*Asb4^−/−^* dams were lower than those in the placentas from NC-*Asb4^−/−^* dams (Figure 4B,C). The mRNA and protein levels of VEGF were not different in maternal adipose tissues from *Asb4^−/−^* dams on different diets (Appendix A). At the time of mating, the plasma levels of VEGF in HFD-*Asb4^−/−^* females were not lower than those in NC-*Asb4^−/−^* females (Figure 4A).

Plasma sFLT1 levels are significantly higher in preeclamptic women than in those with a normal pregnancy. However, in this study, diet had no effects on the plasma sFLT1 levels in *Asb4^−/−^* dams (NC-*Asb4^−/−^*: 1.00 ± 0.18 vs. HFD-*Asb4^−/−^*: 1.12 ± 0.16, *p* = 0.62). Similarly, obesity causes low-degree inflammation, and visceral white adipose tissues can actively synthesize inflammatory markers, including TNFα. In a human study, the expression of TNFα in the visceral adipose tissues from PE patients was higher than that from normal pregnant women [20]. However, in this study, the mRNA levels of *Tnfα* in either visceral adipose tissues or placentas from *Asb4^−/−^* dams on the two diets did not differ (Appendix A).

### 2.5. Inhibitor of DNA-Binding Protein 2 (ID2) Amount in Placentas from HFD-Asb4^−/−^ Dams Significantly Increase

A previous study showed that a subset of trophoblasts remains undifferentiated with sustained high ID2 expression in the placenta of *Asb4^−/−^* dams and that this plays a role in preeclampsia [8]. We, therefore, examined whether HFD feeding affects ID2 expression in the placentas of *Asb4^−/−^* dams. Immunostaining analysis showed that there was stronger ID2 staining (fluorescent intensity of about 1.5×) in the placentas from 12.5 dpc HFD-*Asb4^−/−^* dams than that in the placentas from NC-*Asb4^−/−^* dams (Figure 5A,B). ID2 staining was predominantly located in the labyrinthine region, which was consistent with a previous report [8]. Western blot also showed that the placentas from HFD-*Asb4^−/−^* dams had ~2× ID2 levels than the placentas from NC-*Asb4^−/−^* dams at 18.5 dpc (Figure 5C,D). The mRNA levels of *Id2* in the placentas from HFD-*Asb4^−/−^* dams tended to be higher, but the difference did not reach a significant level (Figure 5E). 

Park et al. reported that *Id2* mRNA levels increase in epididymal adipose tissues in genetic and diet-induced obese mice [21]; however, *Id2* mRNA levels were not higher in the gonadal adipose tissues of HFD-*Asb4^−/−^* dams compared to NC-*Asb4^−/−^* dams (0.76 ± 0.07 in HFD groups relative to 1.00 ± 0.18 in NC groups, *p* = 0.15).

### 2.6. Insulin, but Not Leptin, Increases ID2 in First-Trimester Trophoblasts

We hypothesized that changes induced by obesity in maternal serum factor(s) would causally affect placental development and lead to preeclamptic phenotypes. The process is so dynamic that different factors may have an effect at different stages. As a start, we examined the effects of insulin and leptin on the first-trimester trophoblasts: previously, Takao et al. showed that ID2 is involved in insulin sensitivity and glucose uptake [22]. Furthermore, Palei et al. showed that direct administration of insulin to pregnant animals leads to complications of pregnancy [23]. However, the effects of insulin on ID2 expression have not been studied. Human first-trimester trophoblasts (HTR8/SVneo) express insulin receptor and ID2 [24], but we found that these cells do not express detectable levels of ASB4, using qRT-PCR with two different sets of primers and a probe (Appendix A). We treated HTR8 cells with insulin at doses of 0, 1, and 10 nM for 24 or 48 h in a serum-deprived condition. Insulin tended to increase mRNA and protein levels of ID2 in HTR8 cells, but neither change reached a significant level after 24 h treatment (Figure 6A–C). After 48 h treatment, mRNA levels of *ID2* in treated cells were not different from those in control (0 nM) cells. In contrast, ID2 levels were significantly higher at about 2× in cells treated with 10 nM insulin than in control cells (Figure 6D–F). 

Insulin treatment did not alter the glucose levels in the culture medium (control: 174.7 ± 0.68 mg/dL; 1 nM insulin: 186.1 ± 3.1 mg/dL; 10 nM insulin: 183.2 ± 4.2 mg/dL, *p* = 0.09), which was consistent with the report by Vega et al. [25]. Additionally, Chen et al. reported that the protein expression of VEGF significantly decreases in HUVECs exposed to high levels of insulin (100 nM) [26], while Hale et al. reported that insulin at a dose of 10 nM directly stimulates VEGF production in the glomerular podocyte [27]. However, when we measured the VEGF expression in HTR8 cells after exposure to different doses of insulin, the mRNA level of VEGF was not different among different treatment groups after 24 h (1.00 ± 0.20 in 0 nM, 1.56 ± 0.22 in 1 nM, 1.86 ± 0.99 in 10 nM, *p* = 0.66). VEGF was undetectable in the culture supernatant, even after being approximately 10× concentrated by Amicon^®^Ultra-0.5 Centrifugal Filter 3K Devices. 

As described before, HFD feeding increased the plasma levels of leptin in *Asb4^−/−^* females before mating as well as during pregnancy. Since leptin is reported to be involved in placentation [28], we measured the expression of ID2 in trophoblasts (HTR8) treated with different doses of leptin in vitro. No increase in the expression of ID2 was detectable at the doses tested (Appendix A).

## 3. Discussion

This study demonstrated that HFD-induced obesity leads to increased circulating insulin levels and worsens the preeclampsia-like phenotypes in pregnant mice lacking ASB4, as evidenced by HFD-*Asb4^−/−^* dams having a higher BP and urinary albumin excretion, more severe kidney impairment, and worse pregnant outcomes than NC-*Asb4^−/−^* dams. In addition, the levels of VEGF were lower in HFD-*Asb4^−/−^* dams than in NC-*Asb4^−/−^* dams. A subset of trophoblasts remained undifferentiated in the placentas from NC-*Asb4^−/−^* dams, and HFD feeding further perturbed the differentiation, as judged by the increased ID2 in the placentas from HFD-*Asb4^−/−^* dams. Our in vitro data showed that insulin increases ID2 levels in a dose- and time-dependent manner in human trophoblasts. Collectively, our data suggest that hyperinsulinemia resulting from diet-induced obesity impairs placental vascular differentiation, decreases placental VEGF expression, and aggravates fetal and maternal phenotypes of preeclampsia in *Asb4^−/−^* dams.

Obesity is a risk factor for preeclampsia [4]. However, the precise mechanisms underlying this phenomenon are still unclear. Many factors can be considered to play a role in increasing the risk for preeclampsia in obesity at various stages of pregnancy. Such factors include hyperinsulinemia, inflammation, oxidative stress, and angiogenic imbalance [7]. The effect of HFD on preeclampsia in experimental animals has been investigated by multiple research groups; however, the results are not consistent, in part due to different groups of researchers applying different experimental approaches. For example, Masuyama et al. reported that 12-week-old Institute of Cancer Research (ICR) female WT mice develop obesity and preeclampsia-like phenotypes during pregnancy after 4-week treatment with HFD, having 62% calories from fat [29]. In contrast, Palei et al. surprisingly found that pregnant rats fed an HFD (42% fat kcal) for 9 weeks starting at 12 weeks old are normotensive and, contrary to their initial hypothesis, present enhanced sensitivity to acetylcholine-induced endothelium-dependent vasorelaxation in small arteries and nitric oxide (NO)/endothelial nitric oxide synthase (eNOS) compared to dams fed a normal diet [30]. Sun et al. established preeclampsia-like conditions in pregnant mice by injecting L-arginine-methyl-ester, a NOS inhibitor, at the early and mid-gestation period and reported that an HFD aggravates preeclampsia-like phenotypes and lipid infiltrations in the liver and placenta [31]. To begin investigating the mechanism underlying the effects of obesity on preeclampsia, with a particular focus on the stage of trophoblast differentiation, we chose to study *Asb4^−/−^* females that develop mild preeclampsia-like phenotypes.

During early placentation, high levels of ID2 in trophoblast progenitors ensure that the proliferation reaches the proper numbers of trophoblasts before they differentiate. Once the proper numbers are reached, downregulation of ID2 is required for the progenitors to differentiate. This controlling process is important and well documented. For example, cultured human cytotrophoblasts with constitutively high ID2 expression via an adenovirus vector retained the characteristics of undifferentiated cells, and the invasion of these cells was impaired [32]. Furthermore, Selesniemi et al. demonstrated that increasing or decreasing *Id2* expression in mouse labyrinthine trophoblast progenitor cells (SM10) leads to the prevention or promotion of differentiation induced by TGFβ [33]. Instead, ASB4 is expressed in a specific and narrow spaciotemporal fashion during embryo and placental development, ensures the degradation of ID2, and thereby promotes differentiation into a variety of cells. While ASB4 is unlikely to be the sole molecule involved in the ID2 degradation process, its complete absence delays and impairs the placental differentiation process, leading to preeclampsia [8]. Our demonstration that the placentas from HFD-*Asb4^−/−^* dams had a further increase in the expression of ID2 being sustained to further late stages suggests that the adverse effects of maternal obesity on preeclampsia begin at the trophoblast differentiation stage.

Previous studies have shown that insulin is required for differentiation of stem cells, but excess insulin has direct harmful effects on trophoblasts, including increased DNA damage, apoptosis, and decreased cell survival [25]. However, the effect of higher-than-normal yet still physiological levels of insulin, as in obese individuals, on ID2 expression was unknown. In this study, we stimulated human first-trimester trophoblasts (HTR8) with insulin at a dose that is comparable to the concentration observed in patients with insulin resistance [25,34] and observed that ID2 expression in these cells increased in a dose- and time-dependent fashion. Our result suggests that excess insulin supply through maternal blood may directly inhibit trophoblast differentiation by increasing ID2 levels. 

ASB4 negatively regulates ID2 through polyubiquitination and proteasome-dependent degradation [8]. We found that HTR8 cells do not express ASB4 (Appendix A); therefore, effect of insulin on the accumulation of ID2 must be an ASB4-independent phenomenon in this setting. Consistent with in vitro data, in vivo data show that ID2 expression increases in the placentas from obese dams lacking ASB4 with hyperinsulinemia. While no other specific mechanism has been proposed in ID2 degradation at this point, ASB4 is not likely to be the sole player, although important, since placentation in mice can continue without ASB4. The transcription of ID2 is up- and downregulated by many factors, including BMP9, Ovol2, [35], and type 1 insulin growth factor receptor (IGF1R). Somewhat surprisingly, however, we were not able to detect a significant response to insulin at the mRNA levels of ID2 in HTR8 cells. Insulin is also known to influence protein translation as well as stability [36], and Silva et al. reported that insulin treatment of HTR8 cells is hypertrophic since the treatment increases protein contents without affecting the general transcription levels of genes [37]. We are aware of the limitation of our study using HTR8 cells. Although they possess trophoblast-progenitor-cell-like characteristics [38] and have been well accepted as a model of extra villus trophoblasts, they are an immortalized cell line and consequently must differ in factors for cell survival and apoptosis regulation. Further investigations on the effects of insulin on ID2 expression and protein degradation are necessary.

Finally, we note that ASB4 is expressed in promelanocortin neurons in the hypothalamus, mediates insulin receptor substrate 4 (IRS4) degradation [39,40], and controls satiety and glucose homeostasis [39,41]. Additionally, the association between variants of the ASB4 locus and obesity has been identified in humans [42]. A small but significant increase in the adiposity of *Asb4^−/−^* females and its exaggeration by HFD feeding is likely to result from the absence of ASB4 in these neurons. Future studies must, therefore, include whether neuronal glucose regulation also contributes to the exaggerated insulin increase in the maternal circulation during pregnancy and whether the genetic variants of ASB4 influence preeclampsia. Regardless, variants of ASB4 as well as those in other protein degradation mechanisms that control ID2 levels during trophoblast differentiation require further attention as risk factors for preeclampsia. 

## 4. Materials and Methods

### 4.1. Mice 

WT and *Asb4^−/−^* mice (129/SvEv background) [8,12] were housed in standard cages on a 12 h light/dark cycle with free access to food and water. All experiments were carried out in accordance with the National Institutes of Health guideline for use and care of experimental animals, as approved by the IACUC of the University of North Carolina at Chapel Hill: protocol numbers: 21-235 and 22-229. 

### 4.2. Diet and Mating Strategy 

At time of weaning (21–23 days old), female littermates of WT or *Asb4^−/−^* were randomly enrolled into a normal chow (NC, 14% calories from fat, 3002909-203, PicoLab, Fort Worth, TX, USA) [10,43] or a high fat diet (HFD, 42% calories from fat, TD88137, Harlan Teklad, Madison, WI, USA) [11,44] group. Five weeks later, WT females were mated with WT males, and *Asb4^−/−^* females were mated with *Asb4^−/−^* males. Females were checked for vaginal plugs each morning, and the day of plug detection was designated as 0.5 days post coitus (dpc) [18].

### 4.3. Blood Pressure (BP) Measurements 

At the third wk of pregnancy, BP was measured by the computerized tail-cuff method. Mice were habituated to the machine for one day (13.5 dpc). Thirty measurements were collected for the following 5 days (14.5 to 18.5 dpc). The average BP of the five individual day was used for analysis. All mice were trained on the BP apparatus for 10 cycles of measurements before 30 measurements were made each day as described previously [8,12]. 

### 4.4. Cell Culture 

The HTR8/SVneo (HTR8) trophoblast cell line was kindly provided by Dr. C.H. Graham, Queen’s University, Kingston, Ontario, Canada [45], and maintained in RPMI-1640 medium supplemented with 5% FBS. Cells were starved for 18 h with 0 % FBS, then treated with insulin at doses of 0, 1, 10 nM for 24 h or 48 h [25,37], and another batch of cells were treated with leptin at doses of 0, 5, 50, 500 ng/ml for 24 h [46,47]. After incubation, medium and cells were collected for analysis. Insulin was purchased from Sigma-Aldrich (#91077C, St. Louis, MO, USA); leptin from R&D Systems (#398LP01M). Experiments were repeated two times with each experimental group consisted of 3 replicates

### 4.5. Other Procedures 

Detailed methods for plasma biochemical measurements, urinary albumin, morphological examination, immunofluorescence (IF), Western blot (WB) and antibodies, and quantitative RT-PCR and primer sequences are described in Appendix A. 

### 4.6. Statistical Analysis 

Data are presented as the mean ± SEM. Two-way ANOVA or the *t*-test was performed using the program JMP 16.0 (SAS Institute Inc. Cary, NC, USA). Post hoc analyses were conducted using the Tukey–Kramer honest significant difference test. Differences were considered to be statistically significant with *p* values < 0.05.

## 5. Conclusions

HFD feeding leads to obesity and hyperinsulinemia in mice. Although this treatment regimen does not affect pregnancy in WT mice, pregnant mice lacking ASB4 with hyperinsulinemia show increased placental ID2 and decreased plasma VEGF and develop more severe preeclampsia-like phenotypes. In addition, insulin directly increases ID2 in trophoblasts. Collectively, these results imply that insulin itself directly executes detrimental effects during the early stage of placentation. Interrogation of maternal insulin in preeclampsia may reveal novel pathways to target for prevention.

## Figures and Tables

**Figure 1 ijms-24-02149-f001:**
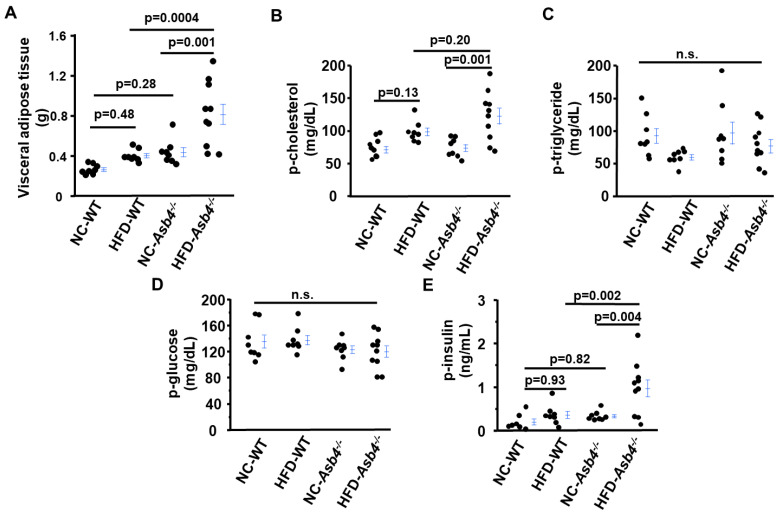
A high-fat diet (HFD) increases the visceral adipose tissue mass and circulating cholesterol and insulin levels in *Asb4^−/−^* dams but not triglyceride and glucose levels. WT and *Asb4^−/−^* pregnant mice were euthanized at 18.5 days post-coitus (dpc) after 4 h fasting, and blood was collected for determining (**A**) visceral adipose tissue mass, (**B**) plasma (p) cholesterol, (**C**) triglyceride, (**D**) glucose, and (**E**) insulin. WT: wild type; NC: normal diet; HFD: high-fat diet; n.s.: not significant. *n* = 8 (NC-WT), 8 (HFD-WT), 8 (NC-*Asb4^−/−^*), 10 (HFD-*Asb4^−/−^*).

**Figure 2 ijms-24-02149-f002:**
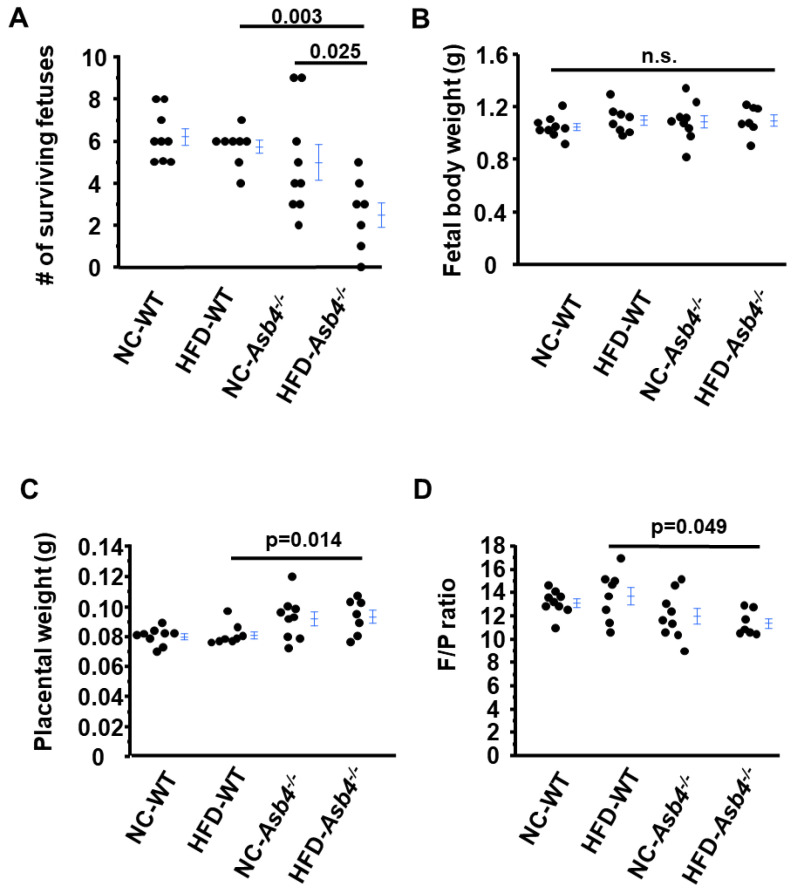
*Asb4^−/−^* pregnant mice fed a high-fat diet (HFD) have decreased surviving fetuses and decreased fetal weight/placental weight ratio. WT and *Asb4^−/−^* pregnant mice were euthanized at 18.5 days post-coitus (dpc). The number of surviving fetuses was counted, and the fetal body weight and placental weight were measured. (**A**) Average number of surviving fetuses. (**B**) Fetal body weight. (**C**) Placental weight. (**D**) Fetal body weight/placental weight (F/P) ratio. NC: normal diet; HFD: high-fat diet; n.s.: not significant. *n* = 9 (NC-WT), 8 (HFD-WT), 9 (NC-*Asb4^−/−^*), 8 (HFD-*Asb4^−/−^*).

**Figure 3 ijms-24-02149-f003:**
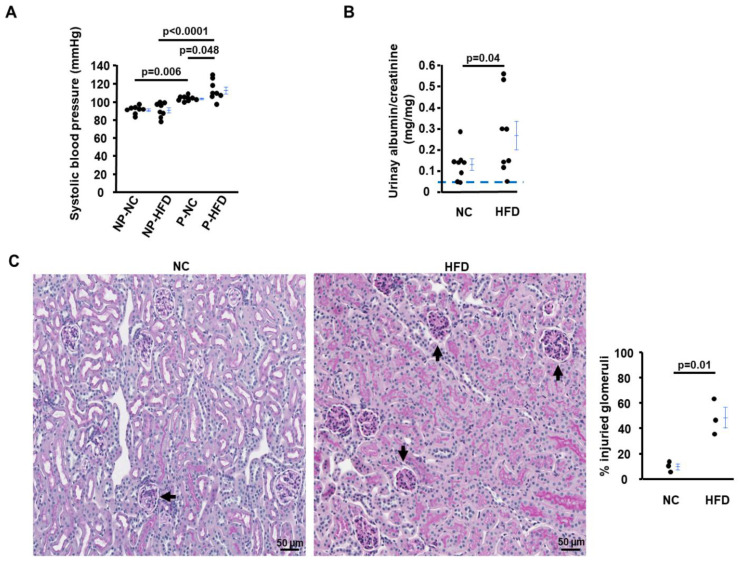
*Asb4^−/−^* pregnant mice fed a high-fat diet (HFD) have higher blood pressure and urinary protein and more kidney damage than *Asb4^−/−^* pregnant mice fed normal chow (NC). (**A**) Systolic blood pressure (SBP) in *Asb4^−/−^* female mice (non-pregnant and pregnant) after approximately 7 weeks of HFD or NC feeding regimen. NP-NC: non-pregnant with NC; NP-HFD: non-pregnant with HFD; P-NC: pregnant mice with NC; P-HFD: pregnant mice with HFD. (**B**) Urinary albumin/creatinine ratio in *Asb4^−/−^* dams at 18.5 days post-coitus (dpc). The blue dashed line indicates the normal value. *n* ≥ 8. (**C**) Representative glomeruli from two different diet groups. Periodic acid–Schiff stain. Right panel: semi-quantitation of the percentage of injured glomeruli; *n* = 3. Black arrow: injured glomeruli.

**Figure 4 ijms-24-02149-f004:**
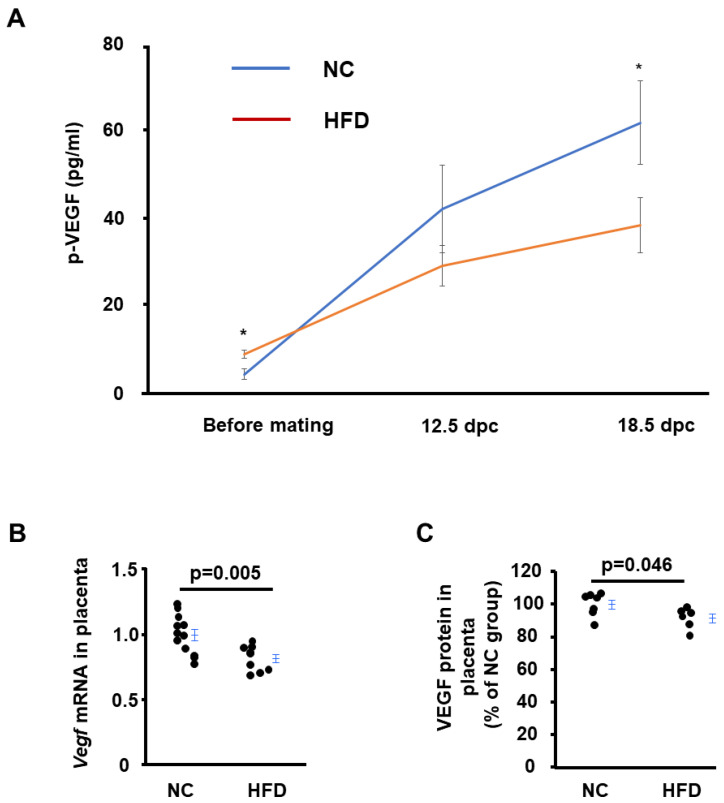
*Asb4^−/−^* pregnant mice fed a high-fat diet (HFD) have decreased VEGF levels in circulation and the placenta. Placentas were collected from *Asb4^−/−^* dams on normal chow (NC) or a high-fat diet (HFD) at 18.5 dpc. (**A**) Plasma levels of VEGF at different stages of pregnancy; p: plasma. * *p* < 0.05. Individual animal values at each time point are shown in Appendix A. (**B**) mRNA levels of *Vegf* in placentas; *n* = 12 (NC), 9 (HFD). (**C**) Protein levels of VEGF in placentas. NC: normal chow; HFD: high-fat diet. *n* = 7 (NC), 6 (HFD).

**Figure 5 ijms-24-02149-f005:**
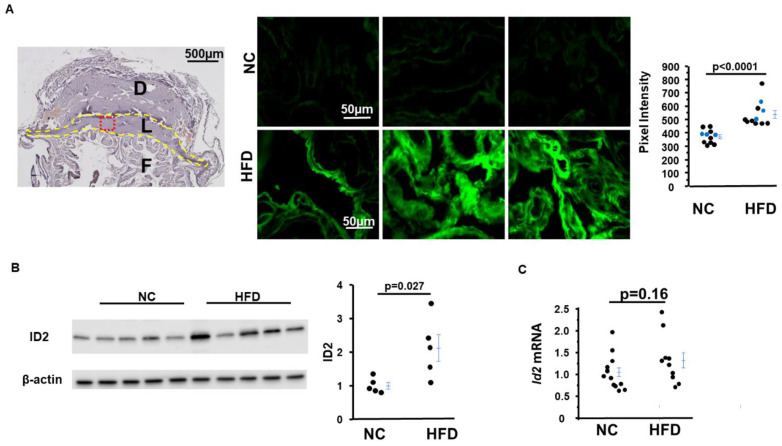
*Asb4^−/−^* pregnant mice fed a high-fat diet (HFD) have increased ID2 in the placenta. (**A**) Left panel: structure of the placenta at 12.5 dpc. The red region is where the pictures were taken. D: decidua; L: labyrinthine; F: fetus. Middle panel: representative of ID2 immunofluorescence images of placentas from *Asb4^−/−^* dams on normal chow (NC) or a high-fat diet (HFD). Each image is from a placenta from an individual dam. Right panel: semi-quantification of the density of fluorescence. Blue dots present the values of the pictures in the middle panels. (**B**) Western blot (left panel) and densitometric quantitation (right panel) of ID2 in the placentas from *Asb4^−/−^* dams on normal chow (NC) or a high-fat diet (HFD) at 18.5 dpc; *n* = 5. (**C**) mRNA of *Id2* in the placentas from *Asb4^−/−^* dams on normal chow (NC) or a high-fat diet (HFD) at 18.5 dpc; *n* ≥ 10.

**Figure 6 ijms-24-02149-f006:**
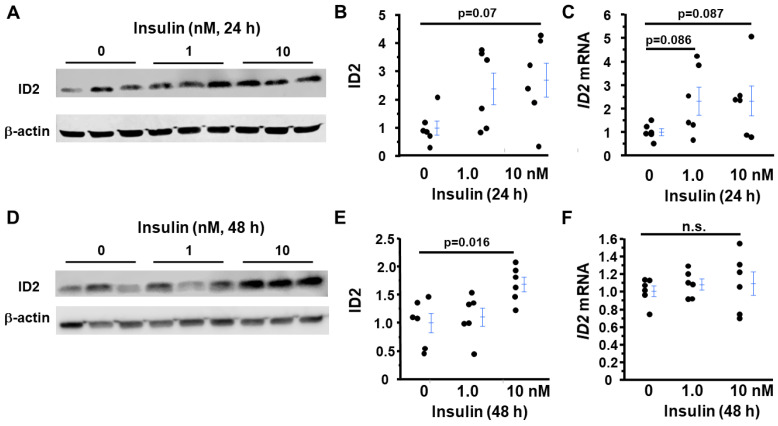
Insulin induces ID2 expression in human trophoblasts. After starving with 0% FBS medium for 18 h, human first-trimester trophoblasts (HTR8/SVneo) were exposed to 0, 1, and 10 nM insulin for 24 h (**A**–**C**) or 48 h (**D**–**F**). (**A**,**D**). Western blot and (**B**,**E**) densitometric quantitation of ID2; *n* = 6. (**C**,**F**) mRNA of *ID2* in cells treated with different doses of insulin; *n* = 6. n.s.: not significant.

## Data Availability

The data supporting the results of this article are included within the article. The data presented in this study are available on request from the corresponding author.

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
