# Peer review of "Insulin Elevates ID2 Expression in Trophoblasts and Aggravates Preeclampsia in Obese ASB4-Null Mice"

_ijms, 2023, doi:10.3390/ijms24032149_

Round 1

Reviewer 1 Report

It is a well conducted study with a testable hypothesis and research question. The authors have demonstrated the role of high fat diet in exacerbating pregnancy outcomes in mice lacking ASB4. This is accompanied by increased ID2 expression caused by increased insulin sensitivity as a reult of hyperinsulinemia leading to preeclampsia like phenotype. The experiments are well designed and the results are presented well.

The manuscript in review by Kayashima et. al. attempts to elucidate the mechanisms that trigger preeclampsia under conditions of obesity. To this effect they study the significance of ankyrin-repeat-and-SOCS-box-containing-protein 4 (ASB4) at an earlier implantation stage of pregnancy, trophoblast differentiation. It is a rather bold hypothesis the authors were successful at demonstrating that ASB4 knock-out mice when obese develop preeclampsia-like symptoms. Further they demonstrate that the obesity lead hyperinsulinemia interferes with 'inhibitor of DNA-binding 2' (ID2), a protein known to disturb trophoblast differentiation.

The experiments are well-designed and the results presented are of superior quality. They confirm their hypothesis and provide substantial evidence indicating the same. The manuscript presented is sound in scientific nature in terms of giving relevant background information, explanation of results and discussion. The research conducted has a high novelty value as to my knowledge it is the first time the role of ASB4 in predisposed obesity is studied during pregnancy, nonetheless at such an early stage of pregnancy. Therefore this will be of great interest to readers. The authors have further tried to explore the mechanism by which ASB4 is involved and depict that the clearance of ID2 protein is perturbed leading to its accumulation, while ID2 gene expression does not change. This clearance perturbation is attributed to hyperinsulinemia induced by obesity. 

Author Response

Thank you for reviewing our manuscript. We appreciate your comments!

Reviewer 2 Report

Dear Authors my comments:

1. Is needed part before abstract?

2. All citations have to have references; in several places are not written references.

3. Part "introduction"- second paragraph- first sentence no chance to understand.

4. Part "materials and methods" no information about number of permission form Ethic Committee for research. 

5. In my opinion references should be written in square brackets "[]" and before "." not after.

Reviewer 3 Report

The objective of the current manuscript was to investigate the role of obesity on preeclampsia using mice lacking the ankyrin-repeated and SOCS box containing protein 4 (ASB4).  The authors showed that ASB4 knockout animals treated with a high-fat diet prior to pregnancy and during pregnancy showed preeclamptic phenotype including an increase in blood pressure and proteinuria and kidney pathology.  They also showed an increase in white fat weight, plasma leptin and insulin levels and increase in ID2 protein levels in the placenta.  The authors concluded that hyperinsulinemia perturbs the removal of ID2 protein and interferes with trophoblast differentiation contributing to enhanced preeclampsia.  The findings are interesting but there several major issues which needs to be addressed before it can be considered for publication.

1)    Blood pressure was collected for 5 days (14.5 days to 18.5 days) and average of 5 days was used for analysis.  It has been shown in mice pregnancy that blood pressure dips down in mid gestation (14.5 and 15.5) and starts to increase in late pregnancy.  So, taking an average blood pressure measurement from 5 time point may lead incorrect blood pressure readings.  Showing blood pressure measurement at the late gestation time point (17.5 and 18.5) will be the most relevant time point to indicate if these animals show preeclampsia like phenotype.

2)     Once of the major findings is that the involvement of ASB4 in trophoblast differentiation.  Therefore, information on trophoblast invasion, spiral artery remodeling, placental vascularizaton in the ASB4 KO placentas during pregnancy is required to understand the role of ASB4 in placental development.

3)     In abstract, explaining what is ASB4 and its role in pregnancy will be useful.  Furthermore, in the abstract the authors state that these animals show preeclampsia like phenotype but stating what the exact phenotype will also be useful.

4)     In Figure 1, the KO animals treated with HFD appears to show a greater variability in parameters such as visceral adipose tissue, p-cholesterol, p-triglyceride, p-glucose and p-insulin parameters. What can account for this greater variability?  Was amount of HFD food intake measured? 

5)     Figure 3-5, WT control data for normal diet and HFD data is missing for most of the parameters including urinary albumin/creatinine levels, % injured glomeruli, p-VEGF, VEGF mRNA in the placenta, VEGF protein levels in the placenta, ID2 protein and mRNA levels etc.  WT data is required for proper control for these experiments.

Round 2

Reviewer 2 Report

Dear Authors,

I accept your reply.